# Identifying opportunities to optimize mass drug administration for soil-transmitted helminths: A visualization and descriptive analysis using process mapping

Eileen Kazura[1‡]*, Jabaselvi Johnson[2‡], Chloe Morozoff[1,3], Kumudha Aruldas[2], Euripide Avokpaho[4], Comlanvi Innocent Togbevi[4], Félicien Chabi[4], Marie-Claire Gwayi-Chore[1,3], Providence Nindi[5], Angelin Titus[2], Parfait Houngbegnon[4], Saravanakumar Puthupalayam Kaliappan[2], Yesudoss Jacob[2], James Simwanza[5], Khumbo Kalua[5], Judd L. Walson[3,6], Moudachirou Ibikounlé[4,7], Sitara S. R. Ajjampur[2], Arianna Rubin Means[1,3]

**1** The Department of Global Health, University of Washington, Seattle, Washington, United States of America, **2** The Wellcome Trust Research Laboratory, Division of Gastrointestinal Sciences, Christian Medical College, Vellore, India, **3** The DeWorm3 Project, Seattle, Washington, United States of America, **4** Institut de Recherche Clinique du Bénin, Abomey-Calavi, Bénin, **5** Blantyre Institute for Community Outreach, Blantyre, Malawi, **6** The Departments of Global Health, Medicine, Pediatrics and Epidemiology, University of Washington, Seattle, Washington, United States of America, **7** Centre de Recherche pour la lutte contre les Maladies Infectieuses Tropicales (CReMIT/TIDRC), Université d'Abomey-Calavi, Benin

‡ These authors share first authorship on this work.
* eakazura@gmail.com

## Abstract

### Background

The control of soil-transmitted helminths (STH) is achieved through mass drug administration (MDA) with deworming medications targeting children and other high-risk groups. Recent evidence suggests that it may be possible to interrupt STH transmission by deworming individuals of all ages via community-wide MDA (cMDA). However, a change in delivery platforms will require altering implementation processes.

### Methods

We used process mapping, an operational research methodology, to describe the activities required for effective implementation of school-based and cMDA in 18 heterogenous areas and over three years in Benin, India, and Malawi. Planned activities were identified during workshops prior to initiation of a large cMDA trial (the DeWorm3 trial). The process maps were updated annually post-implementation, including adding or removing activities (e.g., adaptations) and determining whether activities occurred according to plan. Descriptive analyses were performed to quantify differences and similarities at baseline and over three implementation years. Comparative analyses were also conducted between study sites and areas implementing school-based vs. cMDA. Digitized process maps were developed to provide a visualization of MDA processes and inspected to identify implementation bottlenecks and inefficient activity flows.

**Data Availability Statement:** Data cannot be shared publicly because the study remains blinded to outcome data. Data are available from the

DeWorm3 Institutional Data Access Committee (contact via deworm3@uw.edu, https://depts.washington.edu/deworm3/) for researchers who meet the criteria for access to these data.

**Funding:** The DeWorm3 study is funded through a grant from the Bill and Melinda Gates Foundation (OPP1129535, PI JLW) (https://www.gatesfoundation.org/). The funders had no role in study design, data collection and analysis, decision to publish, or preparation of the manuscript.

**Competing interests:** The authors declare that they have no competing interests.

## Results

Across three years and all clusters, implementation of cMDA required an average of 13 additional distinct activities and was adapted more often (5.2 adaptations per year) than school-based MDA. An average of 41% of activities across both MDA platforms did not occur according to planned timelines; however, deviations were often purposeful to improve implementation efficiency or effectiveness. Visualized process maps demonstrated that receipt of drugs at the local level may be an implementation bottleneck. Many activities rely on the effective setting of MDA dates and estimating quantity of drugs, suggesting that the timing of these activities is important to meet planned programmatic outcomes.

## Conclusion

Implementation processes were heterogenous across settings, suggesting that MDA is highly context and resource dependent and that there are many viable ways to implement MDA. Process mapping could be deployed to support a transition from a school-based control program to community-wide STH transmission interruption program and potentially to enable integration with other community-based campaigns.

## Trial registration

NCT03014167.

### Author summary

Research suggests that many contextual factors affect the success of mass drug administration (MDA) campaigns, yet detailed descriptions of implementation is absent from existing literature. We applied process mapping to describe and compare the flow of activities in MDA programs for soil-transmitted helminths. This represents the first known use of process mapping in community-based campaigns in low resource settings. Process maps were updated annually over three implementation years to identify activities that were added or removed from the MDA process, and whether implementation occurred according to plans. Findings suggest that MDA processes vary widely, are regularly adapted, and deviations from plans often occurred for purposeful reasons. Process mapping may be valuable for microplanning of MDA campaigns because it can generate granular detail about planned activities across national, sub-national and local levels.

## Background

Worldwide, over 1.5 billion people are infected with soil-transmitted helminths (STH) [1,2], a group of intestinal parasites which include *Ascaris lumbricoides*, *Ancylostoma duodenale*, *Necator americanus* and *Trichuris trichiura*. These infections are associated with up to 2 million disability adjusted life years annually [3], and chronic or high-intensity infections are associated with several nutritional and cognitive morbidities, including malnutrition, intestinal complications, anemia, poor growth, preterm birth, and cognitive impairment among children [4,5]. The World Health Organization (WHO) recommends that endemic countries control STH with mass drug administration (MDA) [2,6] of anthelmintic drugs (albendazole or

mebendazole), targeting at least 75% coverage of the highest risk populations [7], including children and women of reproductive age. MDA for STH is typically implemented through school-based delivery platforms [8] and targets pre- and school aged children and non-pregnant adolescent girls and women, due to their high-risk for STH infection and the significant potential benefits of treatment [9]. Though the number of children treated worldwide has increased from about 30 to 90 million in the past decade [10], many STH programs fail to achieve global targets of 75% coverage [11,12], which attenuates the potential impact of MDA programs. Recent evidence suggests broadening deworming treatment to all age groups via community-wide MDA (cMDA) may interrupt STH transmission in targeted geographic areas [13–15]. Moreover, cMDA has been shown to be an equitable delivery platform capable of reaching the most vulnerable and underprivileged subpopulations [14].

Planning for MDA implementation for STH includes estimating the number of people who are at risk for infection, procuring and transporting deworming drugs to the community level, and administering drugs to people in their homes or at schools. Evidence suggests that common barriers to achieving high MDA coverage in schools or communities include lack of adequate community sensitization about the campaign and the safety of deworming drugs, inadequate human resources to deliver drugs, limited training for drug distributors, and a lack of coordination between national or regional program managers and local communities [16–19]. This suggests that multiple infrastructural and contextual factors including *how* MDA is implemented impacts coverage and, ultimately, disease transmission.

Transitioning to a policy of cMDA for STH transmission interruption will require new approaches to MDA delivery, as high coverage will be fundamental for achieving elimination. Process mapping (PM) provides an opportunity to identify activities necessary for delivery of cMDA with high coverage. This can inform microplanning such as that described in the WHO's recently released microplanning guide which indicates the importance of "plan[ning] the activities required to reach and treat target populations" [20]. PM is a systems science tool by which implementers inventory the activities or steps necessary for successful implementation and map the flow, or process, of these steps [21,22]. PM has been used extensively in the health sector to study the flow of patients through health care settings [23–25] and in quality improvement efforts for outpatient procedures, leading to gains in efficiency or safety [26,27]. PM can help create shared understanding of workflows and responsibilities, identify potential gaps, bottlenecks and accelerators in implementation processes, and assist with adaptation of implementation protocols to local context [21,28]. Thus, PM of MDA across contexts can provide evidence about the influence of context on implementation.

We analyzed PM data from the DeWorm3 study, a hybrid type I cluster randomized trial in Benin, India, and Malawi testing the feasibility of interrupting STH transmission via biannual cMDA [29,30]. The purpose of this analysis is to describe the processes that select clusters in the DeWorm3 study followed to implement school-based distribution and cMDA over three years, and to describe the differences and similarities in implementation across delivery platforms and settings. Findings from this multi-country analysis may inform creation of a future tool to guide context-adapted implementation of cMDA for STH.

## Methods

### Ethics statement

The DeWorm3 study was reviewed and approved by the Institut de Recherche Clinique au Bénin (IRCB) through the National Ethics Committee for Health Research (002-2017/CNERS-MS) from the Ministry of Health in Benin, The London School of Hygiene and Tropical Medicine (12013), The College of Medicine Research Ethics Committee (P.04/17/2161) in

Malawi, and the Institutional Review Board at Christian Medical College, Vellore (10392). The DeWorm3 Project was also approved by The Human Subjects Division at the University of Washington (STUDY00000180). The trial was registered at ClinicalTrials.gov: NCT03014167.

## Study sites

The DeWorm3 study is ongoing in the Commune of Comé, Benin, Timiri and Jawadhu Hills communities in Tamil Nadu, India, and Mangochi District in Malawi. The DeWorm3 trial rationale and study design are described in detail elsewhere [29,30]. In total, over 370,000 individuals are included in the DeWorm3 trial population, with each DeWorm3 site including a baseline population ranging from 94,969 to 140,932 (censused between October 2017—February 2018). Each site was divided into 40 clusters, with approximately 2–3,000 individuals per cluster (ranging from 1650 to 4000). In each site, 20 clusters were randomized to the intervention (biannual cMDA) and 20 clusters were randomized to be controls and follow the country's standard of care school-based deworming program.

## Cluster sampling

Within each site, PM was conducted in six clusters: four intervention clusters implementing cMDA and two control clusters implementing standard-of-care school-based MDA (N = 18 clusters total). Clusters were selected using a stratified, randomized process based on historical treatment coverage; clusters that successfully implemented MDA with high coverage in the past may be more likely to successfully implement MDA in DeWorm3. Previous implementation units (the smallest geographic level of MDA implementation) were identified as having either historically high (over 80%) or low (below 60%) coverage of school-based MDA, then randomly selected to ensure that half of sampled clusters were historically high and half low coverage. The closest overlapping DeWorm3 clusters to the selected implementation units were included PM data collection.

## Data collection

Prior to the first round of cMDA in 2018, stakeholders from the DeWorm3 study, the Ministry of Health and/or Education, and partner organization staff familiar with MDA planning and implementation participated in PM workshops at the cluster level (N = 18 workshops) from February to June of 2018. Each workshop followed a standardized guide to identify all activities considered necessary for delivering MDA with high coverage in the given cluster (S1 Text). During the workshop, each activity was classified into one of seven pre-determined categories: drug supply chain, community sensitization, training, planning, MDA delivery, monitoring and evaluation (M&E), or other. For each activity, workshop participants identified the ideal goal and timeline for completion of the activity. Timing categories included up to 2 months before MDA, 2 months to 2 weeks before MDA, 2 weeks before the start of MDA, during MDA, or after MDA. For example, an activity such as 'contact village leaders to notify them of the upcoming MDA campaign', might include a target of '100% of village leaders notified' and a target timeline of 'one month prior to commencing MDA'. Participants also developed a visual process map to diagram the flow of activities over time and determine relationships between activities (e.g., identification of drug distributors occurs before training of drug distributors). PM workshops used a participatory, nominal group technique to reach consensus and produce a shared vision of implementation activities.

Baseline process maps were updated once annually for three years. During one round of MDA in years 1, 2 and 3, observed progress towards activity goals and timelines were tracked in real-time for all activities. After MDA, each cluster updated their process maps, reporting

on implementation of all previously identified activities. Each PM update also recorded reasons for any deviation from the goals and timelines observed. In addition, new activities could be added and activities could be removed from implementation plans during annual updates.

### Data analysis

The characteristics of baseline process maps were compared across sites, delivery method (school-based or cMDA), historical coverage level, and timing category. The range and average number of activities per cluster and the proportion in each activity category were calculated. Chi-square tests ($\alpha = 0.05$) were performed to identify differences in the distribution of activity categories between delivery platforms and historical coverage levels, and whether cMDA had a larger proportion of community sensitization activities than school-based MDA. Additionally, the number and proportion of adaptations, activities that were added or removed from the implementation plan, were calculated over time. For example, adding a new community sensitization activity, such as street plays to inform the community of upcoming MDA, was considered an adaptation.

In order to assess fidelity to implementation plans, the total number of deviations from the baseline maps and annual updates were also calculated. Deviations were defined as differences between implemented and planned activity goals or timelines. For example, an activity that took place ahead of planned timelines or occurred later than planned would be considered a deviation. Thus, deviations are not necessarily negative reflections of implementation but may also reflect implementation realities. Responses to open-ended question soliciting the reason for each deviation were categorized into one or more of eight deviation options, including: purposeful changes to increase efficiency, purposeful changes to increase effectiveness, competing priorities or dependency delays, community influences, resource constraints, linkages with Ministry of Health programs, COVID-19, or another reason. For instance, if an activity was delayed due to a later than expected decision on MDA dates by the national program, the reason was attributed to Ministry of Health linkages. The proportion of activities that deviated from goals and timelines were reported separately for each PM update. During the first PM update, the two school-based clusters in Malawi updated data late and these data were excluded from the analysis to reduce the risk of recall bias. MDA treatment coverage was uniformly high across all clusters and rounds, and potential associations between coverage and adaptations or deviations will be reported in future manuscripts.

Cluster-level process maps were digitized using the DiagrammeR package [31] and RStudio [32] to depict all mapped activities, including those that were added or removed over the course of the trial, and visualize changes in implementation over time.

### Results

This study analyzed PM data from 18 clusters across three countries, and over three years of MDA implementation. The analysis indicates that there is a high degree of variability in implementation processes across countries and across clusters implementing cMDA and school-based MDA and little variation between areas with historically high and low treatment coverage (S1–S3 Tables).

An active adaptation of activities occurred between baseline planning and the first (year 1) update, though fine tuning of implementation processes persisted over time. While deviations from plans were common, many deviations represented changes in the timing of delivery and many deviations were purposeful to increase the efficiency or effectiveness of implementation.

## Implementation plans

Eighteen clusters conducted PM workshops to describe plans for implementing school-based MDA or cMDA (Table 1). Clusters in India identified a larger number of activities needed to achieve high coverage (average of 71.3), as compared with Benin (average of 27.5) and Malawi (average of 30.5). A larger proportion (38.6%) of identified activities in India were planning activities, as compared to the other two sites. Across all sites, clusters implementing cMDA had a larger number of activities (average of 47.5) than clusters implementing school-based MDA (average of 34.3). Chi-square tests found no difference in the distribution of activities across categories between school-based and cMDA ($\chi^2 = 9.7$, df = 6, p = 0.14) and historically high and low coverage clusters ($\chi^2 = 9.9$, df = 6, p = 0.13). cMDA included a statistically significant ($\chi^2 = 6.4$, df = 1, p = 0.01) larger proportion of community sensitization activities (average of 19.8%) than school-based MDA (average of 11.7%).

Planning and community sensitization activities were scheduled to take place mainly in the three pre-MDA time categories. Drug delivery activities mainly took place during MDA

**Table 1. Characteristics of MDA implementation plans prior to the first round of MDA.**

| | Total activities[1] | | Proportion of activities, by category (%) | | | | | | |
|---|---|---|---|---|---|---|---|---|---|
| | Average per cluster | Total (range) | Planning | Drug Supply Chain | Training | Community Sensitization | MDA Delivery | Monitoring & Evaluation | Other |
| **Country** | | | | | | | | | |
| Benin | 27.5 | 165 (19–32) | 13.9% | 24.2% | 19.4% | 20.0% | 15.8% | 4.8% | 1.8% |
| India | 71.3 | 428 (49–91) | 38.6% | 26.4% | 7.2% | 15.2% | 10.7% | 1.6% | 0.2% |
| Malawi | 30.5 | 183 (27–36) | 19.7% | 16.9% | 8.2% | 21.3% | 14.2% | 19.1% | 0.5% |
| **Intervention[2]** | | | | | | | | | |
| School-based | 34.3 | 206 (19–53) | 30.6% | 24.3% | 10.7% | 11.7% | 13.1% | 9.2% | 0.5% |
| Community-wide | 47.5 | 570 (27–91) | 28.2% | 23.5% | 9.8% | 19.8% | 12.5% | 5.4% | 0.7% |
| **Historical coverage[3]** | | | | | | | | | |
| Low coverage | 42.6 | 383 (26–81) | 25.6% | 26.6% | 11.7% | 17.8% | 11.7% | 6.3% | 0.3% |
| High coverage | 43.7 | 393 (19–91) | 32.1% | 20.9% | 8.4% | 17.6% | 13.5% | 6.6% | 1.0% |
| **Activity timing** | | | | | | | | | |
| Up to 2 months before MDA | 4.4 | 79 (1–20) | 28 (35.4%) | 33 (41.8%) | 5 (6.3%) | 13 (16.5%) | 0 (0.0%) | 0 (0.0%) | 0 (0.0%) |
| 2 months– 2 weeks before MDA | 16.7 | 300 (5–47) | 141 (47.0%) | 61 (20.3%) | 37 (12.3%) | 57 (19.0%) | 4 (1.3%) | 0 (0.0%) | 0 (0.0%) |
| 2 weeks–beginning of MDA | 11.2 | 201 (2–34) | 43 (21.4%) | 57 (28.4%) | 36 (17.9%) | 63 (31.3%) | 1 (0.5%) | 1 (0.5%) | 0 (0.0%) |
| During MDA | 6.1 | 110 (2–12) | 11 (10.0%) | 7 (6.4%) | 0 (0.0%) | 4 (3.6%) | 78 (70.9%) | 8 (7.3%) | 2 (1.8%) |
| After MDA | 4.8 | 86 (2–12) | 1 (1.2%) | 26 (30.2%) | 0 (0.0%) | 0 (0.0%) | 15 (17.4%) | 41 (47.7%) | 3 (3.5%) |

[1]Average is total number of activities/total number of relevant clusters; range is defined as the minimum and maximum number of activities across the relevant clusters

[2]School-based MDA includes data from 6 clusters and cMDA includes data from 12 clusters

[3]Historically high coverage clusters had over 80% coverage; historically low coverage had below 60% coverage

(70.9%) and the after-MDA period primarily included drug supply chain (30.2%) and M&E (47.7%) activities. Drug supply chain activities occur throughout the pre- and post-MDA periods, reflecting activities related to ordering, acquiring, and transporting drugs before MDA, and collecting and storing drugs after MDA has finished. Cluster-level implementation plans are available in S2 Table.

## Adaptations to plans

Adaptations are defined as activities that are added or removed over the course of three years of implementation. Most adaptations were reported during the first PM update and an average of 5.2 adaptations to cMDA were made at each update, while an average of 2.7 adaptations to school-based MDA were made per update (Table 2). Implementers continued to adapt plans over three years and across activity types, though training activities were adapted least. In Benin, the maps generally expanded as more activities were added than were removed. In India and Malawi, the maps generally shrunk as activities were removed from the plans made at baseline. For example, in India, initial plans indicated a drug request letter would be sent from the state ministry of health (MOH) to national MOH, however this activity was removed because the drugs were allotted directly from the national MOH and collected by the Deworm3 implementation partner. Adaptation metrics for each cluster are detailed in S2 Table.

**Table 2. Adaptations to MDA implementation processes over time.**

|  | Total adaptations[1] | | Percent change in number of activities from previous year | | |
|---|---|---|---|---|---|
|  | Average per year | Total over three years (range)[2] | Year 1 | Year 2 | Year 3 |
| **Country** | | | | | |
| Benin | 4.2 | 76 (3–18) | 21.8% | 5.4% | 4.2% |
| India | 4.1 | 74 (6–20) | -9.1% | -2.8% | 0.0% |
| Malawi | 4.7 | 85 (10–18) | -17.5% | 12.6% | -1.1% |
| **Intervention[3]** | | | | | |
| School-based | 2.7 | 49 (3–14) | -4.4% | -2.5% | -1.6% |
| Community-wide | 5.2 | 186 (7–20) | -4.6% | 4.4% | 1.8% |
| **Time** | | | | | |
| Up to 2 months before MDA | 0.4 | 22 (0–8) | -20.3% | 0.0% | 0.0% |
| 2 months– 2 weeks before MDA | 1.2 | 65 (0–8) | -7.3% | -3.6% | 2.6% |
| 2 weeks–beginning of MDA | 1.1 | 59 (0–8) | 2.5% | 3.9% | 1.9% |
| During MDA | 0.7 | 40 (0–8) | 10.9% | 9% | 0.7% |
| After MDA | 0.9 | 49 (0–7) | -16.5% | 5.6% | -6.7% |
| **Activity Categories** | | | | | |
| Planning | 0.9 | 46 (0–9) | -7.1% | -8.2% | -0.5% |
| Drug Supply Chain | 0.8 | 45 (0–5) | -9.9% | 9.8% | 0.6% |
| Training | 0.3 | 17 (0–4) | 7.7% | 1.2% | -2.4% |
| Community Sensitization | 0.9 | 47 (0–6) | -1.5% | 6.7% | 0.0% |
| MDA Delivery | 0.9 | 48 (0–11) | 16.3% | 7.9% | 8.9% |
| M&E | 0.5 | 29 (0–6) | -40.0% | 6.7% | -9.4% |
| Other | 0.1 | 3 (0–2) | -20.0% | -25.0% | 33.3% |

[1]Number of activities either added or removed

[2]Minimum–maximum number of adaptations by a cluster in a single round of MDA

[3]School-based MDA includes data from 6 clusters and cMDA includes data from 12 clusters

During the first PM update (year 1), most adaptations involved stopping activities that took place more than two months before MDA (20.3% decline) and post-MDA activities (16.5% decline). Many of these activities were one time, start-up activities, such as performing a census, or resulted from changes to the drug supply chain or reporting requirements. In contrast, the number of activities performed in the two weeks leading up to MDA (2.5% growth) and during MDA (10.9% growth) increased during the first PM update, mainly due to new additions from Benin. Benin included the least number of activities on average during baseline workshops, and thus new activities were added as implementation plans were solidified. The addition of activities in the two weeks preceding MDA and during MDA continued in the year 2 update (3.9% and 13.9% growth).

During the year 1 update, 21 activities in Malawi had been significantly changed. Activities that were significantly changed are not recorded as adaptations in Table 2, as they were not removed but significantly altered to adjust for local context. These changes were made in each of the six clusters in Malawi and were attributable to changes in responsibilities for drug supply chain and monitoring and evaluation activities. For example, implementation partners were responsible for the receipt and storage of study drugs at the local level, instead of the public health system as originally planned at baseline. Similar changes were made in Malawi due to the introduction of electronic tools for reporting coverage.

While adaptations at the year 1 update were largely focused on refinement of processes, adaptations made in years 2 and 3 included responses to contextual factors, such as extreme weather events and the COVID-19 pandemic. Across the clusters, numerous activities were added to increase coverage and assist with ascertainment of coverage. In addition to adding mop-up activities, multiple clusters also added outreach to people who initially refused drugs, and supervision of drug distribution during MDA. Activities such as thumb marking, mid-line assessment of MDA delivery, and documentation of remaining drugs were added to improve coverage estimates. Most clusters also refined their community sensitization strategies over the course of the three years. For example, in India, all the intervention clusters added street plays as an additional activity in year 2 to increase community awareness.

## Fidelity to plans

PM data provides insight into two aspects of implementation fidelity: adherence to planned implementation timelines and goals. Fidelity, the extent to which activities were implemented as planned, varied across countries and years, and there was higher fidelity to activity goals than timelines (see S4 Table for deviation data across all countries and time periods). In each country, 30–57% of activities were not executed according to planned timelines (e.g. had deviations) (Fig 1). There were fewer deviations from planned implementation goals, though deviations increased in year 3, attributed to the COVID-19 pandemic. Timeline fidelity was highest for activities which took place during MDA (average 0.3 deviations per cluster per year).

As with adaptations, fidelity varied across the three study sites. Though cMDA for STH was only recently introduced and underwent significant adaptation, goal and timeline fidelity were similar between school-based MDA and cMDA (S3 Table). Clusters with historically high and low coverage also exhibited similar fidelity to implementation plans.

Deviations are neither inherently positive nor negative. In fact, many deviations in this study were intentional to increase implementation efficiency or effectiveness. Many training activities, for instance, occurred later than planned to increase information retention leading into MDA delivery, and some planning and drug supply chain activities occurred earlier than planned to reduce implementer workloads closer to MDA. The climb in purposeful deviations

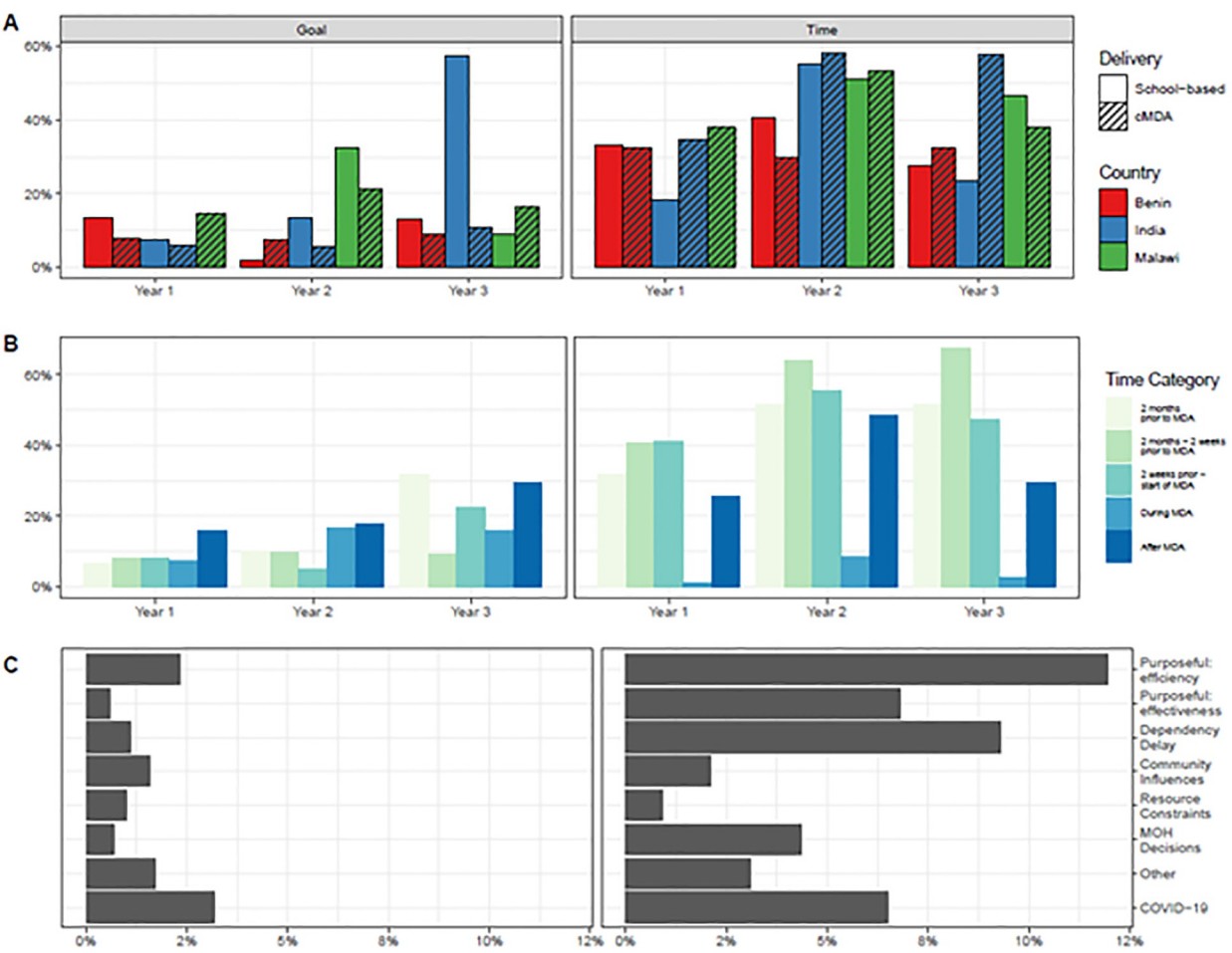

**Fig 1. Deviations from planned activity goals and timelines.** Panel A: Percent of activities with goal (left) and time (right) deviations, by delivery platform and country. Panel B: Percent of activities with goal (left) and time (right) deviations, by activity timing category. Panel C: Percent of activities with goal (left) and time (right) deviations over three years attributable to different deviation categories.

to increase efficiency observed during the year 2 update (18.3%, see S3 Table) were due to having systems and supplies already in place from the previous rounds of MDA. For example, drug distributors were already recruited, and medical kits already compiled to address adverse events. Goal and time deviation metrics for each cluster are presented in S4 Table.

### Process map visualizations

Visualizing process maps can help identify potential inefficiencies or bottlenecks in MDA implementation, points of coordination between sectors and implementation levels (local, regional, national), and viable adaptations. Fig 2 shows the process map for Cluster L in Benin, which implemented cMDA (process maps for all 18 clusters can be found at https://rpubs. com/ekazura/pm-viz). The visualization includes activities (color coded by activity category), time categories, activities which were added after baseline PM workshops, and activities which were removed from implementation plans. The map demonstrates that, in this cluster, three key drug supply chain activities were clustered within the two weeks prior to MDA. Because these activities cascade from one another, this highlights a potential point of dependency

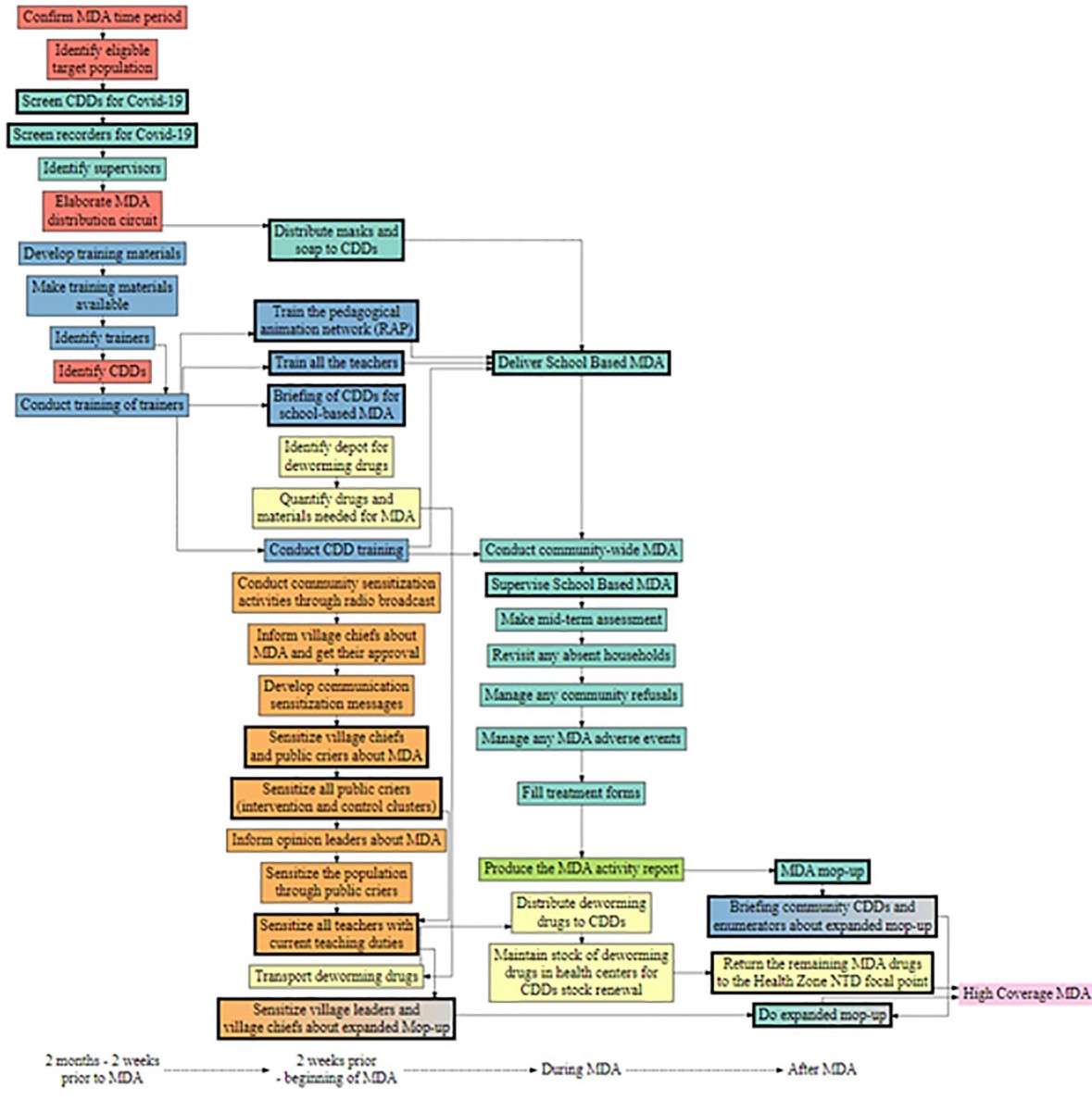

**Fig 2. Process map, Cluster L, Benin.**

delay. By identifying the depot for deworming drugs or quantifying the number of drugs needed further in advance of the launch of cMDA, delays in one activity may be less likely to cause bottlenecks for other implementation activities.

The map also highlights that multiple activities were added after baseline PM workshops. These adaptations were due to the integration of some school-based drug delivery activities into cMDA implementation plans, the COVID-19 pandemic, and a weather event which caused a need for expanded mop-up activities. For example, some training and community sensitization activities for school-based delivery were added into the cMDA cascade when cMDA implementers assumed responsibility for drug delivery at schools as well.

There are several commonalities that are visually evident within the 18 digitized maps. First, early activities in all cluster cascades include estimating the amounts of drugs needed

and setting dates for MDA. In most clusters, there are a number of intermediary steps between receipt of drugs at the local or regional level and the point of providing drugs to field workers for distribution. Examples include crushing drugs for young children, documenting and storing drugs safely and checking expiration dates. The process maps also illustrate the variation in drug supply chain activities across clusters and highlight receipt of the drugs at the relevant local level as a potential bottleneck because it is a key node from which multiple other activities cascade.

## Discussion

In this study, longitudinal PM data provide a highly detailed portrait of school-based MDA and cMDA implementation processes across 18 clusters in multiple settings. This includes characterizing the number and type of planned activities, adaptations, fidelity to implementation, and the activity cascades (i.e., activity sequencing) used to implement MDA. We found that drug supply chain activities spanned the pre- during- and post-MDA timeframes, planning activities were most common, and that the cMDA platform implemented more community sensitization activities than school-based delivery. An active adaptation period occurred after the first round of cMDA, though implementers continued to refine processes over the three years. The duration of the MDA activity cascade shortened over time for both MDA platforms; activities were most often removed from the pre- and post-MDA periods and added during MDA. Finally, we found a high degree of deviations from planned timelines, many for purposeful reasons to increase efficiency or effectiveness of implementation.

Planning activities were most common in both school-based and cMDA clusters, and while the number of planning activities decreased modestly over time, in year three about a quarter of all activities were still allocated for planning. This indicates the importance of planning for both delivery platforms, particularly as preemptive investments in planning may lessen the impact of commonly cited challenges to attaining high coverage, such as a lack of standard practices and plans [33], a lack of staff or volunteers to deliver drugs [34], and inadequate supervision during MDA [35].

Community sensitization is critical to achieving high coverage [18,36–38], and we found that cMDA included more community sensitization activities than school-based delivery. A transition from school-based delivery to cMDA may necessitate a multifaceted community sensitization approach to inform community members of upcoming MDA and encourage compliance.

For both school-based distribution and cMDA, drug supply chain activities occur throughout the pre- and post-MDA periods. This finding validates recently released WHO supply chain guidance [39] and provides additional detail on how drug supply chain activities interface with other types of activities, such as training or drug delivery. For example, PM data from India revealed intermediary steps, such as crushing drugs for young children and sending drugs for quality checks, between receipt of drugs at the regional level and distribution to the local level, not included in the guidance document. In this way, PM can track how implementers 'adopt, adapt or contextualize [40] global drug supply chain guidance to fit local context and campaign implementation plans. In addition, the highly detailed view of the implementation cascade produced by PM may be used to identify areas of possible integration with other community-based campaigns. For example, understanding the steps in the drug supply chain for STH deworming may allow planners to realistically select other campaigns or commodities that could share supply chain timelines and resources.

Results also indicate that adaptation (removal or addition of activities) to implementation processes occurred throughout the three-year study period. This suggests that MDA is

inherently dynamic and implementation plans should not be considered static documents but rather must necessarily adapt to the changing context. This aligns with guidance developed for immunization [41] and trachoma [42] campaigns, which suggests that an iterative planning approach (i.e. microplanning) supports pre-emptive problem solving and can help strengthen health systems. Clusters implementing cMDA reported a higher number of adaptations than those implementing school-based MDA at their first update; because school-based MDA is established in all settings, the program had likely already undergone an active adaptation period prior to PM monitoring. Additionally, cMDA reaches more people with a more intensive door-to-door delivery method, instead of fixed-point delivery through the school system, which may be a more complex process requiring additional adaptations to achieve coverage goals. Building organizational cultures that normalize mid-course adjustments and iterative planning [43] may be particularly well suited to settings transitioning from a school-based to community-wide campaign approaches.

In addition to adaptations made at each round of MDA, implementation timelines generally condensed as activities that took place far in advance of and after MDA were most likely to be removed, and activities were most commonly added during MDA itself. Many of the added activities, such as supervision of drug distributors and counseling of people who initially refuse drugs, are activities specifically designed to increase coverage. In this study, map visualizations across all 18 clusters suggests that estimating drug quantities, setting dates for MDA and receipt of drugs at the local level could be potential bottlenecks to delivery of MDA with high coverage. This suggests a need for close supervision or other supportive adaptions to ensure timely completion of these activities.

Implementation fidelity is often critical for achieving intended intervention outcomes [44–46] in community-based and resource constrained settings [47–49]. In this study, many activities deviated from planned implementation timelines, though often these deviations were driven by purposeful reasons, either to create efficiencies or to implement MDA more effectively. Less frequently, deviations were due to dependency delays resulting from arrival of drugs to the local level, late setting of MDA dates by the national program, and late completion of household lists used by drug distributors to track coverage. This demonstrates the importance of both flexibility in MDA implementation and careful tracking of activities to quickly address observed challenges.

Fidelity to implementation timelines was highest for activities occurring during MDA as opposed to before and after MDA. It may be that the time-bound nature of drug delivery during MDA necessitates high fidelity, or that activities which occur during MDA are most important to deliver as planned, so implementers are less likely to deviate from these plans. Future implementation research embedded within DeWorm3 will focus on identification of core activities which require fidelity in order to achieve high coverage MDA.

While commonly used in quality improvement projects and hospital-based settings, to our knowledge this is one of the first applications of PM in a community-based healthcare campaign in a LMIC setting. Tracking of activity times, goals, and reasons for deviations illustrates that fidelity is multi-faceted and deviations from plans not inherently negative. Limitations to comparing activity cascades across settings include the potential for the same activity to be named or conceptualized differently from setting to setting. For example, while one cluster indicated "planning for training" as one activity, another may have broken that down into multiple, more specific activities. However, multiple rounds of quality checks and iterative name standardization meetings were conducted to ensure consistency in activity conception and categorization across sites during the data cleaning phase of this study. Also, in this study, PM was not used as a quality improvement tool to test potential adaptations to increase coverage. While we believe there is potential for PM to be integrated into microplanning in this

way, this study does not explore its effectiveness as a tool to increase coverage. Finally, PM took place within the DeWorm3 trial environment, where study staff supported PM activities and collection of updated data. PM may need to be simplified to match resource constraints of a particular national or sub-national campaign if used routinely in the future.

## Conclusion

In this study, PM was used to collect data about the specific flow of activities needed to successfully implement MDA, how that activity flow changed over time, and fidelity to activity goals and timelines. Visualizations of the activity flow were created to broker a shared understanding of the implementation process among stakeholders and to identify areas of potential implementation bottlenecks. PM is a low-technology, easy to learn tool which can be used as an entry point for microplanning efforts, a guide for process improvement over time, and to identify potential areas for synergy and integration with other campaigns. PM can also be used across geographies to generate and share new knowledge of adaptations and innovations as they arise. Across the 18 clusters included in this study, large variation in activity flows were observed, suggesting that MDA is highly context and resource dependent and that there are many viable ways to implement MDA depending upon the setting.

## Supporting information

**S1 Text. In-depth process mapping activity worksheet.** Worksheet used by clusters to identify activities for process maps and set ideal goal and timelines.
(DOCX)

**S1 Table. Characteristics of MDA implementation plans for each cluster, prior to the first round of MDA.** Proportion of activities in each activity category, by cluster.
(DOCX)

**S2 Table. Adaptations to MDA implementation processes for each cluster.** Total adaptations and percent change in number of activities compared to previous round for each cluster.
(DOCX)

**S3 Table. Deviations from planned goals and timelines.** Average deviations per round and proportion of activities with goal and time deviations by a number of cluster and activity characteristics.
(DOCX)

**S4 Table. Goal and time deviations in each round for each cluster.** Average goal and time deviations and proportion of activities with goal and time deviations at each update, for each cluster.
(DOCX)

**S1 Checklist. StaRI checklist for reporting of implementation science studies.**
(DOCX)

## Author Contributions

**Conceptualization:** Marie-Claire Gwayi-Chore, Judd L. Walson, Arianna Rubin Means.

**Data curation:** Eileen Kazura, Jabaselvi Johnson, Chloe Morozoff, Kumudha Aruldas, Euripide Avokpaho, James Simwanza.

**Formal analysis:** Eileen Kazura.

**Funding acquisition:** Judd L. Walson.

**Methodology:** Eileen Kazura, Chloe Morozoff, Arianna Rubin Means.

**Supervision:** Khumbo Kalua, Judd L. Walson, Moudachirou Ibikounlé, Sitara S. R. Ajjampur, Arianna Rubin Means.

**Validation:** Jabaselvi Johnson.

**Visualization:** Eileen Kazura.

**Writing – original draft:** Eileen Kazura, Chloe Morozoff.

**Writing – review & editing:** Jabaselvi Johnson, Kumudha Aruldas, Euripide Avokpaho, Comlanvi Innocent Togbevi, Félicien Chabi, Marie-Claire Gwayi-Chore, Providence Nindi, Angelin Titus, Parfait Houngbegnon, Saravanakumar Puthupalayam Kaliappan, Yesudoss Jacob, James Simwanza, Khumbo Kalua, Judd L. Walson, Moudachirou Ibikounlé, Sitara S. R. Ajjampur, Arianna Rubin Means.

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
