## [Decision Letter · Decision Letter 0]

21 Apr 2023

Dear Ms Kazura,

Thank you very much for submitting your manuscript "Identifying opportunities to optimize mass drug administration for soil-transmitted helminths: a visualization and descriptive analysis using process mapping" for consideration at PLOS Neglected Tropical Diseases. As with all papers reviewed by the journal, your manuscript was reviewed by members of the editorial board and by several independent reviewers. The reviewers appreciated the attention to an important topic. Based on the reviews, we are likely to accept this manuscript for publication, providing that you modify the manuscript according to the review recommendations. 

Sincerely,

Uwem Friday Ekpo, PhD

Academic Editor

Francesca Tamarozzi

Section Editor

Reviewer's Responses to Questions

**Key Review Criteria Required for Acceptance?**

**Methods**

-Are the objectives of the study clearly articulated with a clear testable hypothesis stated?

-Is the study design appropriate to address the stated objectives?

-Is the population clearly described and appropriate for the hypothesis being tested?

-Is the sample size sufficient to ensure adequate power to address the hypothesis being tested?

-Were correct statistical analysis used to support conclusions?

-Are there concerns about ethical or regulatory requirements being met?

Reviewer #1: (No Response)

Reviewer #2: The methods met all the required conditions

**Results**

-Does the analysis presented match the analysis plan?

-Are the results clearly and completely presented?

-Are the figures (Tables, Images) of sufficient quality for clarity?

Reviewer #1: (No Response)

Reviewer #2: Yes, The results are clear and completely presented, but some suggestions have been made. Also, the quality of the figures needs to be improved upon.

**Conclusions**

-Are the conclusions supported by the data presented?

-Are the limitations of analysis clearly described?

-Do the authors discuss how these data can be helpful to advance our understanding of the topic under study?

-Is public health relevance addressed?

Reviewer #1: (No Response)

Reviewer #2: The conclusions are supported by data presented, limitations well stated and public health relevance clearly highlighted throughout the manuscript.

**Editorial and Data Presentation Modifications?**

Reviewer #1: (No Response)

Reviewer #2: (No Response)

**Summary and General Comments**

Reviewer #1: This is an interesting and welcome paper. Highlighting that even when activities are not highly technical, they can and do vary significantly between contexts: “MDA is highly context and resource dependent and that there are many viable ways to implement MDA” (lines 51-52). It’s a concept which is often overlooked when we aim for low-cost / ease of implementation.

Process Mapping is agnostic to disease, so could be used beyond STHs and beyond NTDs. I note that other similar approaches are available, such as Activity Sequencing, Critical Path Diagram, and even a Gantt chart. So much of what is included in this approach is rooted in common-sense high quality program management. That’s not meant as a guarded criticism, it’s still a necessary step to identify and explore and make explicit.

We would welcome “a future tool to guide context-adapted implementation of cMDA for STH.” (lines 117-118).

Comments

• I understand focus on high and low coverage. However, many clusters will fall into the moderate category (60-80%). Do you think they will differ at all? Do you have an estimate for what proportion of clusters this represents.

• I agree with the emphasis that deviations are not necessarily negative and changes can be purposeful. In general, they can be a sign of good adaptive management. Was / is it possible to collect information as to whether changes were intentional or forced? Likewise, would it be possible to record whether changes were viewed as having a positive or negative impact?

• I think this is a very important concept: “Building organizational cultures that normalize mid-course adjustments and iterative planning(42) may be particularly well suited to settings transitioning from a school-based to community-wide campaign approaches.” To this end, would it be possible to collect information on whether individuals were comfortable suggesting / implementing changes if they felt it would improve the program, rather than slavishly sticking to the plan? 

• You report that the number of planning activities remains high even in Year 3. Could this also reflect high turnover of personnel?

• Why do you think there were so many more activities in India than other countries? Is this related to how they are named / described, or something deeper?

• Figure 2 – Process Map. Many of these activities will likely be happening concurrently, rather than in a neat step-wise manner. Is PM able to capture that?

• You mention the importance of microplanning. Of course, the WHO have just released an NTD Microplanning Guide and training modules – are there explicit links to this in that guide?

• In many countries we are approaching / aiming for the end of standalone STH programs, and rather having them mainstreamed into MOH / government health systems. Can the same process be used in those situations? It’s referenced in lines 385-389 with respect to supply chain specifically, but also thinking more broadly than that.

Reviewer #2: Manuscript Title:

Identifying opportunities to optimize mass drug administration for soil-transmitted

helminths: a visualization and descriptive analysis using process mapping

Reviewer’s decision

The manuscript is of high quality considering the context, relevance, amount of work-done, analysis made, and style of writing/presentation. However, I think it would be more beneficial to have the therapeutic coverage (the % and category) tabulated against the table on adaptation. This answers programmatic questions such as; were the adaptations improving coverage of medicines across the years. This line can also be captured in the result and discussion section (if there is a trend). This should also be performed for fidelity (i.e., deviations).

Reviewer’s Comment

TOPIC:

The title of this manuscript is appropriate and concise, 

INTRODUCTION 

Well detailed, sufficiently referenced, and rationale clearly spelt out

Materials and Methods 

Line 124: authors should recast this line: and study design and implementation science research

RESULTS

Table 1; Authors should also update table 1 with all the chi-square values and p-values, just as they have done within the text. Also, this result is missing on the table “and historically high and

low coverage clusters (�����9.9��df = 6, p = 0.13)…..

Figure 1: This figure currently masks a lot of information (% of deviations). I suggest authors converts this to a table and include coverage estimates as stated in my general comments above.

Discussion:

Line 438: please replace “to try to” with “to”

PLOS authors have the option to publish the peer review history of their article (what does this mean?). If published, this will include your full peer review and any attached files.

Reviewer #1: Yes: Michael French

Reviewer #2: Yes: HAMMED OLADEJI MOGAJI

Figure Files:

Data Requirements:

Reproducibility:

References

---

## [Decision Letter · Decision Letter 1]

6 Nov 2023

Dear Ms Kazura,

We are pleased to inform you that your manuscript 'Identifying opportunities to optimize mass drug administration for soil-transmitted helminths: a visualization and descriptive analysis using process mapping' has been provisionally accepted for publication in PLOS Neglected Tropical Diseases.

Best regards,

Uwem Friday Ekpo, PhD

Academic Editor

Francesca Tamarozzi

Section Editor

Reviewer's Responses to Questions

**Key Review Criteria Required for Acceptance?**

**Methods**

-Are the objectives of the study clearly articulated with a clear testable hypothesis stated?

-Is the study design appropriate to address the stated objectives?

-Is the population clearly described and appropriate for the hypothesis being tested?

-Is the sample size sufficient to ensure adequate power to address the hypothesis being tested?

-Were correct statistical analysis used to support conclusions?

-Are there concerns about ethical or regulatory requirements being met?

Reviewer #1: (No Response)

Reviewer #2: Yes, the methods sections met all the required standards

**Results**

-Does the analysis presented match the analysis plan?

-Are the results clearly and completely presented?

-Are the figures (Tables, Images) of sufficient quality for clarity?

Reviewer #1: (No Response)

Reviewer #2: The results section also has been adequately presented. The concerns raised in the last round of review have been addressed

**Conclusions**

-Are the conclusions supported by the data presented?

-Are the limitations of analysis clearly described?

-Do the authors discuss how these data can be helpful to advance our understanding of the topic under study?

-Is public health relevance addressed?

Reviewer #1: (No Response)

Reviewer #2: Yes. Conclusions aligns with results obtained.

**Editorial and Data Presentation Modifications?**

Reviewer #1: (No Response)

Reviewer #2: (No Response)

**Summary and General Comments**

Reviewer #1: Thank you for the thoughtful responses to comments. Good luck with publication

Reviewer #2: This manuscript is of high quality, which reflects high level experience from authors. I encourage the authors to provide very high quality figures during the production stage

PLOS authors have the option to publish the peer review history of their article (what does this mean?). If published, this will include your full peer review and any attached files.

Reviewer #1: **Yes: **Michael French

Reviewer #2: **Yes: **Mogaji Hammed

---

## [Editor Report · Acceptance letter]

24 Nov 2023

Dear Ms Kazura,

We are delighted to inform you that your manuscript, "Identifying opportunities to optimize mass drug administration for soil-transmitted helminths: a visualization and descriptive analysis using process mapping," has been formally accepted for publication in PLOS Neglected Tropical Diseases.

Best regards,

Shaden Kamhawi

co-Editor-in-Chief

Paul Brindley

co-Editor-in-Chief
